# Qualitative study of the roles of midwives in the provision of sexual and reproductive healthcare services in the Somaliland health system

Rahel Tesfa Maregn  ,[1] Kirsty Bourret,[2,3] Jama Ali Egal,[1,4] Amina Esse,[4] Cristina Mattison,[2,3] Marie Klingberg-Allvin[1,2]

[1]School of Health and Welfare, Dalarna University, Falun, Sweden
[2]Department of Women and Children's Health, Karolinska Institute, Stockholm, Sweden
[3]Department of Obstetrics and Gynecology, McMaster University, Hamilton, Ontario, Canada
[4]Faculty of Nursing and Midwifery, College of Medicine and Health Sciences, University of Hargeisa, Hargeisa, Somalia

**Correspondence to**
Dr Rahel Tesfa Maregn;
h20rahem@du.se and
Professor Marie Klingberg-Allvin;
mkl@du.se

## ABSTRACT

**Objectives** To explore midwives' perspectives in providing sexual reproductive healthcare services in the Somaliland health system.

**Methods** An exploratory qualitative design using focus group discussions (n=6) was used. The study was conducted in the capital of Somaliland, Hargeisa, at six maternal and child healthcare centres that provide sexual and reproductive healthcare (SRH) services. Qualified midwives (n=44) who had been working in the maternal and child health centres for a minimum of 1 year were recruited to participate, and only one did not participate due to illness.

**Results** The results showed that Somaliland midwives face multiple challenges from a lack of formal arrangements, primarily written guidelines and policies, that explicitly define their role as healthcare professionals, which impact the quality of care they provide. They also reported feeling unsafe when practising according to their professional scope of practice due to challenging cultural norms, customary traditions and Somaliland's legal system. Finally, the midwives called for support, including training, institutional protection and psychological support, to enhance their ability and fulfil their role in SRH services in Somaliland.

**Conclusion** Midwives are essential to the provision of equitable SRH services to women and girls, yet are not fully supported by policies, laws or institutions, often living in fear of the consequences of their behaviours. Our research highlights the importance of understanding the context of Somaliland midwifery in order to better support the development of the midwifery workforce, stronger governance structures and midwifery leadership. Appropriately addressing these challenges faced by midwives can better sustain the profession and help to improve the quality of care provided to women and girls and ultimately enhance their reproductive health outcomes.

## STRENGTHS AND LIMITATIONS OF THIS STUDY

⇒ Data were collected by two professional Somali researchers with midwifery backgrounds, coupled with a collaborative approach throughout the research process between the research team in and outside of Somaliland.

⇒ The study participants were frontline workers with first-hand knowledge and experience of the occurrence of the phenomenon. These first-hand accounts provide rich insights into understanding the social and cultural context in which the provision of SRH services takes place.

⇒ While there was much sharing during the focus group discussions, the setting may have limited how much individuals shared. Some may not have wanted to disclose personal details of their experiences in a group setting, especially given the sensitive topic areas.

## INTRODUCTION

Sexual and reproductive health and rights (SRHR) interventions benefit women, girls, children and societies at large. They are considered a cornerstone to achieving universal health coverage and reaching the Sustainable Development Goals.[1 2] Following a life-course approach, the minimum standards of an essential package of SRHR care services include contraceptive counselling; induced abortion, postabortion care (PAC), contraception, antenatal, intrapartum and postnatal care; and detection and prevention of sexual and gender-based violence (SGBV).[2 3] The provision of these essential SRHR care services would contribute to reducing the global maternal mortality ratio to fewer than 70 per 100 000 live births and attaining the target of universal access to SRHR, as set out in the 2030 Agenda for Sustainable Development.[4] Midwives who are educated and regulated according to international standards are an essential part of the health system and can meet 87% of the population's SRHR needs.[5] Midwife-led care, supported by an enabling environment, could save an estimated 4.3 million lives annually by 2035.[6] While scientific evidence shows that midwives are vital to the delivery of safe

and effective SRHR care services, there are significant challenges in the provision of these services within and across countries due to social and cultural factors, gender inequalities and a lack of understanding of the role of midwifery in health systems.[3 7–9]

Somaliland has a high annual maternal mortality rate, at an estimated 396 deaths per 100 000 live births.[10] The leading causes of maternal deaths are postpartum haemorrhage, pre-eclampsia/eclampsia, prolonged and obstructed labour and infections combined with limited access to SRH services.[10–12] The recent Somaliland Health and Demographic Report (2020) showed that 67% of children are delivered at home, further contributing to high maternal and neonatal mortality and morbidity. Moreover, cultural practices such as female genital mutilation/cutting (FGM/C) are prevalent. Approximately 98% of women and girls between the ages of 15–49 years have undergone the procedure, 61% of whom underwent Pharaonic circumcision, a severe form of FGM/C. Access to contraceptives is limited, with 7% of married women reporting practising birth spacing and only 1% use modern contraceptives.[10]

The limited availability of healthcare facilities and the extreme shortage of qualified healthcare professionals further impact health outcomes.[10 11 13] Despite these shortfalls, midwifery institutions and universities in Somaliland follow the International Confederation of Midwives (ICM) standards for midwifery education to meet global requirements and improve the quality of SRHR.[13 14] There are 1313 midwives in Somaliland (993 with diploma certificates, 270 with bachelor's degrees and 50 with master's degrees in midwifery) (Somaliland Nursing and Midwifery Association, 2021). Due to the shortage of healthcare professionals and midwives, low-resource settings like Somaliland often struggle with providing pregnant women with complete healthcare coverage. As a result, women living in these settings rely on non-professional traditional birth attendants who have acquired skills by assisting mothers during home-based childbirths without any medical training.[10–12 15]

While there has been some research on the barriers to specific SRH services in Somaliland, to our knowledge, there is no research that explores the specific challenges and opportunities faced by midwives in this context. This includes understanding the barriers to providing SRH services, such as cultural and social norms, and the strategies midwives use to overcome these barriers. Therefore, our study aims to contribute to a better understanding of midwives' roles in providing SRH services in the Somaliland health system. By exploring the experiences and perspectives of midwives in this context, we can identify the challenges and opportunities they face and inform the development of policies and programmes that support their efforts to provide essential SRH services. This paper is useful for policymakers, practitioners and researchers working in SRHR in Somaliland and similar contexts, as the findings can inform future research and contribute to the development of evidence-based policies

and programmes to support and strengthen midwives' capacity to provide SRHR care services to their full potential. These further improve access to and quality of SRHR care services and ultimately improve overall health outcomes.

## METHODS
We followed an exploratory qualitative study design and used focus group discussions (FGDs) with midwives for data collection.[16 17] The data were analysed inductively using thematic analysis.[18] Since this study is the first of its kind in Somaliland, this design allowed us to understand and describe the nature of the problem.

### Data collection
The study was conducted in the capital of Somaliland, Hargeisa, at six maternal and child health centres (MCHs). MCHs are public facilities monitored by the Ministry of Health. MCHs will either receive clients directly or if they are referred from primary healthcare units, and they must provide basic emergency obstetric and newborn care. The MCHs should be staffed by at least one midwife, a nurse, an auxiliary nurse and a community midwife and cover a population of approximately 5000 people. Six MCHs were targeted for data collection because they served displaced people and host communities, were of equal size and the care was mainly provided by midwives.

A purposive sampling approach was used to recruit practising midwives from the six MCHs. Qualified midwives who met the inclusion criteria had been working in the MCHs for a minimum of 1 year and had experience with SRH service provision. A total of 44 midwives were recruited to participate via face-to-face invitations, and only one did not show up due to illness. The participants were given oral and written information about the study before consent. Six FGDs were conducted in the respective MCHs between January and February 2021. The research team designed an FGD topic guide to facilitate discussion and aid in data collection (see online supplemental appendix 1).[16 17] The guide was piloted in one of the MCHs with seven participants, after which a few probing questions were added. All the FGDs were facilitated by JAE and assisted by AE, both with midwifery backgrounds. Except for one FGD that was conducted in English (RTM was present online), all were conducted in person in the Somali language. Each FGD took approximately 1½ hours.

### Analysis
The audio recordings were transcribed and translated into English by a Somali translator. The transcribed data were checked for accuracy (JAE) and deidentified. In the first step, the entire team carefully read the transcripts of the FGDs multiple times to immerse and become completely familiar with them. The data were then inserted into NVivo V.12, where the analysis team (RTM, KB and CM) coded the data using Braun and Clarke's thematic

**Table 1** Generic codes, Subthemes and Themes showing the data analysis process

| Themes | Subthemes | Generic codes |
|---|---|---|
| Somaliland context and its impact on midwives | Sustaining SRHR care services in the face of resource constraints | Service improvisation with available resources<br>Fundraising<br>Counselling and referral<br>Lack of midwives<br>Lack of service guideline<br>Need for in-service training<br>Service provision in unsafe working environments<br>Need for occupational support |
| | Professional, cultural, and legal dilemmas | Husbands' consent for family planning and post-abortion care<br>Unsafe abortion practices<br>Acting outside the scope of practice<br>Treats and pressures from the client's family members<br>Pregnancy outside marriage and stigma<br>Sociocultural norms behind FGM/C<br>Psychological traumas related to lived experiences of FGM/C<br>Families dealing with sexual violence internally |
| Midwives' perceptions of their role in broader SRHR | Provision of family planning and post-abortion care | Responsible for providing family planning and PAC<br>Premedical assessment<br>Counselling and couples' mediation<br>Breastfeeding guidance as a way of child spacing<br>Referrals and liaising |
| | FGM/C and sexual and gender-based violence | Treating post-FGM/C complications<br>Educating clients about FGM/C<br>Community awareness<br>Using opportunities<br>Providing care for survivors of sexual and gender-based violence<br>Providing care to women with pregnancy outside of marriage |
| | Midwives as advocates and system navigators for clients | Being a role model in using services<br>Use of religious scripts to support women<br>Use female and male support groups for community reach out<br>Women's educational and empowerment programs |

FGM/C, female genital mutilation/cutting; PAC, postabortion care; SRH, sexual and reproductive health; SRHR, sexual and reproductive health and rights.

analysis approach.[18] Codes were reviewed and discussed with the research team (JAE, AE and MK-A). Based on patterns and commonalities, similar codes were grouped as subthemes and themes (RTM, KB and CM) and subsequently reviewed and revised multiple times until the research group reached an agreement that reflected the content of the data (see table 1). Feedback was obtained from the participants after sharing the results (AE and JAE).

### Patient and public involvement

None.

### RESULTS

In the analysis, two main themes were identified: (1) the Somaliland context and its impact on midwives (ie, midwives' working environment at the intersection of health, sociocultural and legal systems) and (2) midwives' perceptions of their role in broader SRHR in the context of Somaliland. Each of these included related subthemes. Context-specific terminologies, such as child spacing/ birth spacing (defined as family planning or contraceptive use) and illegal pregnancy (means the pregnancy has been conceived outside of marriage), were used. The following subthemes elaborate on how the midwives described the lack of occupational and legal support in their work environments and the tensions they experienced between sociocultural norms and their professional role.

### Somaliland context and its impact on midwives

In this theme, midwives describe how the health, sociocultural and legal systems in Somaliland impacted them while providing sexual reproductive health and rights services.

### Sustaining SRHR care services in the face of resource constraints

The midwives stated that there are frequent disruptions in medical equipment and pharmacological supplies and a lack of emergency transport.

> We give women all the family planning information, unfortunately, we do not provide family planning in

this MCH. Those who need family planning, we connect/transfer to other MCHs. (Focus Group 1)

A lack of contraceptives was seen as particularly detrimental, leaving many women exposed to the risk of unintended pregnancy. Furthermore, a lack of supplies meant that the midwives could not manage emergencies, and a lack of transport meant that instead of treating women, they became logisticians, using their cars or collecting funds for taxi fees to escort women to referral hospitals.

We don't have a scanner in the MCH, so we have no way of assessing the level of care a woman needs here […]. If a woman is bleeding severely, we ensure that there is access to a cannula, and we transfer them to the hospital using the team leader's personal car, as the MCH has no ambulance, which we really need. (Focus Group 5)

According to the participants, challenges in service delivery arrangements, such as unclear interprofessional roles and responsibilities, high workloads due to a shortage of midwives and overcrowded hospitals, severely limit the quality of care and contribute to adverse outcomes.

There is a shortage of midwives in Somaliland hospitals […] they have a lot of tasks that occupy their time, overcrowding of people in the hospitals[…] Midwives don't make enough money to sustain their lives, so a lot of them are working double jobs and shifts. There is no monitoring or evaluation from the Ministry as well. (Focus group 6)

The participants claimed that the lack of access to training and hands-on supervision impacted their confidence in providing SRHR service to their clients within the scope of their professional duties. For example, not all midwives have been trained in PAC or feel equipped to implement life-saving skills when faced with complications from incomplete abortions.

We feel it is our job as midwives to help save mothers when they experience early pregnancy complications, but it is sad we cannot. (Focus Group 5)

We, midwives, need more training and experience. We would have a lot of impacts. (Focus Group 6)

They also mentioned a lack of care guidelines, definitions of tasks and legal and occupational support.

There is no legal framework [concerning family planning services] in Somaliland. This lack means healthcare workers do not get the support they need to support women. (Focus Group 5)

### Professional, cultural and legal dilemmas

The participants described how the blending of boundaries between cultural norms, customary traditions and Somaliland's legal system shapes their decision-making and, ultimately, the care pathways of their clients. In Somaliland, there are currently no laws either allowing or prohibiting women from consenting to their own care management. However, cultural norms dictate that either a husband or a male family member must consent to the care, even in urgent or life-threatening circumstances such as an incomplete abortion.

When a woman comes in with abortion complications, you can't help her without her husband. You need the husband present because you need him to sign the consent form, so you don't face any problems afterwards because anything is possible. (Focus Group 6)

Some husbands would, without a doubt, come to us carrying a gun if they ever found out we gave their wives contraceptives without their consent […] but legally, there is no law against giving any method to a woman. (Focus Group 3)

The midwives felt they were professionally and ethically required to provide services to their clients, yet both the midwife and the client could face serious consequences if they did not seek male consent. For example, contrary to cultural norms, it is within the regulatory framework to provide contraception to women without male consent. However, midwives reported being put in unsafe positions if they did so. Verbal, physical, psychological and legal threats from clients' family members were common occurrences in their day-to-day practices.

In the morning, the husband came into the MCHs with a big wooden stick and said, 'Bring out the women in the staff; she will die, or I will.' The police were called, and the husband told the police personnel that the midwife had stopped the childbearing of his wife. There is no law that can guarantee or protect healthcare staff from issues like these. (Focus Group 1)

Unfortunately, we as health care professionals have no law to protect us against a family dispute or if the husband is unhappy about the decision. (Focus Group 3)

These discussions were the most prominent regarding the provision of family planning, emergency PAC and supporting survivors of SGBV.

FGM/C is common practice in Somaliland and participants linked it to sociocultural beliefs around gender norms that include purity, prevention of pregnancy outside marriage and marriage acceptance.

The women believe that all girls who had been performed Sunni (in comparison to the Pharaonic type) are doing an illegal sexual activity, and they are doing naughty things that are illegal […]We (midwives) met many girls who had been circumcised and they still have illegal pregnancy. (Focus Group 2)

Most mothers believe that their daughters will not be accepted for marriage if they don't have a Pharaonic

circumcision. This also includes most men. They say the girl is not pure. (Focus Group 5)

Woven into their narratives were how the midwives were psychologically and emotionally impacted by their own experiences with FGM/C when caring for women who had undergone FGM/C, which is a common occurrence given the extent of the practice.

The woman who cut me really traumatized me. Even years after being cut, I remember getting goosebumps all over my body whenever I saw her. I had ten people holding me down while I was being cut. Now that I know better, I can say what she did to me was completely destructive and damaging. Even to this day, it bothers me to see someone in pain during childbirth. I was basically someone who got surgery without anaesthesia. I was being cut up, and it is still very traumatic. (Focus Group 3)

In terms of psychological distress, all midwives in Somaliland are women, of which most have undergone FGM/C. Participants said some midwives continue to suffer trauma from their own lived experiences with FGM/C. They described how some midwives experience recurring traumatisation in their day-to-day work through treating and supporting women and young girls with complications of FGM/C. They also described a lack of psychological support after providing services to these women and girls. The midwives highlighted the importance of receiving mental health support to better care for themselves and their clients.

We, as midwives, need psychological support to deal with our own trauma; only then can we support women. (Focus Group 3)

Midwives described how sociocultural expectations regarding FGM/C led to ethically difficult scenarios, which at times led to acting outside of their scope of practice. The midwives viewed these instances as a harm-reduction approach to the complications and long-term effects of FGM/C. The midwives also claimed that there is a shift in the demand for FGM/C services from community circumcisers to MCHs and there is mounting social pressure on midwives to provide these services. Often, families cite the risks of turning to unsafe community circumcisers to coerce midwives into providing FGM/C.

A woman said, 'I need circumcisions […] If you refuse, I will get someone else to come in and do it for me.' I refused, and she tried to give me money to do that. (Focus Group 1)

The midwives recognised that refusing to provide FGM/C would inevitably result in them having to treat girls with complications after they were taken to community circumcisers. The risk to the girls, combined with the midwives' own trauma, compelled them to act, even going as far as performing Sunni circumcisions (described by the focus groups as cutting off the tip of the clitoris).

They comply as a way of protecting the girls from a more severe form of FGM/C and related complications.

The reason they perform the Sunni type is to prevent girls from undergoing the Pharaonic type. (Focus group 2)

On the other hand, midwives expressed feeling that the community saw them as playing contradictory roles by performing FGM/C yet promoting its decline.

How is it possible to encourage the community to stop FGM and, on the other hand, we continue the Sunni type? (Focus Group 2)

In the discussions, some midwives questioned the cultural norms behind FGM/C and its practice in the MCHs.

Where does this idea of making a girl 'halaal' (pure) even come from? […] Well, if midwives are given rules to follow, they all need to obey those rules. So, rules need to be created. Holding midwives accountable and having consequences for their actions will make them listen. (Focus Group 3)

Midwives in all six focus groups discussed how the complex intersections of their personal experiences, professional roles, cultural norms and legal framework impacted their personal and professional roles and their capacity to provide quality care to women and girls

### Midwives' perceptions of their role in broader SRHR

When midwives were asked about PAC and contraception, discussions organically led to FGM/C, SGBV and how they advocated for women's SRHR. The three subthemes are discussed below.

### Provision of family planning and post-abortion care

The midwives said that they remained as the primary care providers for women needing PAC and family planning.

Family planning is part of our services, and we tell every mother that comes to the health facility about child spacing. (Focus Group 2)

Midwives have the main responsibility as we see women in pregnancy, labour, and postnatally. (Focus Group 3)

The extent of family planning services provided by midwives includes premedical assessment, family planning counselling, couples' mediation, nutritional support, breastfeeding guidance and liaising and referrals based on women's needs. The midwives provided family planning methods if they were available. While most participants mentioned that it is their responsibility to provide PAC, only in one of the FGDs did the participants mention their competence and willingness to provide PAC.

We do post-abortion care; we got training[…]we use misoprostol. (Focus Group 1)

## FGM/C and sexual and gender-based violence

As mentioned in the previous theme, midwives discussed how the practice of FGM/C impacted their roles as both midwives and women. They stated that their role included providing treatment for emergency and long-term complications of FGM/C, which includes treatment of post-FGM/C bleeding, infection, routine intrapartum care, psychological support and SRHR education relating to FGM/C.

> As a midwife, I tell my clients the best way to prevent illegal behaviour is to educate your children in a proper way. (Focus Group 2)

The midwives supported reducing the FGM/C practice overall through community awareness and educating mothers about the cultural misconceptions and consequences of FGM/C. They explained that their role as educators and awareness creators brought them satisfaction, especially with regard to their personal experience with FGM/C.

> When we have mothers come to us about other issues, like breastfeeding, we use these opportunities to educate them about the issues and problems FGM can cause. (Focus Group 3)

Midwives provide frontline care to survivors of SGBV and women who get pregnant outside of marriage. The participants said they provided equal services without prejudice and gave examples of counselling women and girls who got pregnant outside of marriage.

> Every week we come across two to three cases of rape, and most of those are kids being raped by family members[…] All women have the same options, regardless of where they come from. (Focus Group 6)

## Midwives as advocates and system navigators for clients

The midwives' gender intersected with their professional roles, and some participants mentioned seeing themselves as role models and using their status to support other women. For example, they highlighted the benefits of SRHR services to women and encouraged them to become role models in their respective communities. They also believed that midwives' uptake of services could impact social norms and society, in general, to be more accepting and open to change.

> We all used it and advised each other about it (family planning). (Focus Group 4)

> After the training is finished, they (fathers and mothers' groups) go around villages and look for people with problems. They work with us now and are so valuable and powerful in the community. (Focus Group 1)

In addition to supporting women and each other professionally, they used religious scripts and involved trusted community members in forming male and female support groups, which proved useful for community reach out and social change.

> We are lucky that we have religious evidence that encourages family spacing […] We always use this to our advantage to support mothers. (Focus Group 5)

In some cases, the midwives mentioned acting as advocates for rape survivors and supporting victims to prosecute their perpetrators.

> We asked for an investigation and requested laws and regulations. In this specific case, we ended up winning, getting everyone off our backs, and finally sentencing the guy to jail for 20 years. (Focus Group 6)

The midwives also created programmes to teach women about their reproductive rights and create women's health and social awareness within their communities. This awareness strategy was used to empower women and increase their access to SRHR care services.

## DISCUSSION

To our knowledge, this is the first study to explore the role of midwives in the provision of SRH services in Somaliland and from the perspectives of midwives. Participants described their role expanding beyond antenatal and delivery care to primarily include PAC, contraceptive counselling, SGBV, FGM/C and counselling concerning unintended pregnancies (ie, pregnancies out of wedlock). They also described how they navigate legal, cultural and social spheres, finding creative ways to mitigate and circumvent obstacles for women and girls in dire circumstances. Somaliland midwives face multiple challenges from a lack of formal arrangements, primarily written guidelines and policies, that explicitly define their role as healthcare professionals, leaving them vulnerable to arbitrary persecution and other social consequences. They felt a lack of clinical guidance when managing intrapartum care for women with FGM/C. Furthermore, their lived experiences as women and survivors of FGM/C caused them to experience trauma and suffering in their day-to-day life with little or no mental health support. Under extreme mental stress and lack of safety, midwives are in a constant dilemma. Their decision-making process intersects with their professional, community, religious and individual selves.

Similar studies from different contexts have also shown how midwives improvise strategies to provide emergency SRH care when there is a lack of resources.[19 20] In the absence of clear guidelines and policies, healthcare professionals tend to employ their individual interpretations of social norms and laws.[21] Our results showed no consensus regarding appropriate strategies to address FGM/C practices. Some midwives attempted to deter communities from this sociocultural norm by citing the sexual and reproductive health risks it poses to women and girls. Others argued that FGM/C should be performed in legal facilities to reduce adverse outcomes

of unsafe procedures conducted by non-medical individuals outside clinical settings. Some felt coerced and forced to agree to families' requests. Similar studies in Somaliland and elsewhere have also highlighted the controversies surrounding FGM/C, showing a lack of reconciliation between the two sides of the FGM/C practice spectrum.[22 23]

Our results provide further insight into how midwives' experiences of working with cases of FGM/C and their lived experiences in the Somaliland community may come into play, causing distress and influencing clinical decision-making. This includes navigating the system to provide locally stigmatised SRH services such as PAC, services to women with out-of-wedlock pregnancies and support survivors of SGBV. Midwifery-led interventions, such as contraceptive counselling, are associated with the efficient use of resources and a positive impact on women's reproductive health outcomes.[6] Recent scientific evidence from similar settings shows midwives could play an essential role in PAC treating first and second-trimester incomplete abortion using misoprostol.[24 25] Reports from Somaliland have highlighted that to increase women's access to evidence-based SRHR, midwives must be supported within the health system to broaden their scope of practice in correspondence to ICM Global Standards. The 2020 Somaliland Nursing and Midwifery Association report posits that midwives contribute to improving overall population health and empowering individuals and communities.[26] Midwives in our study clearly state that to continue their work, they require the resources to manage the complexity of the experiences women and girls in Somaliland face regularly. They asked that they urgently receive trauma-informed mental healthcare so they can provide trauma-informed care to women, girls and their communities. While our results highlight the positive impact of midwifery-led care for women and girls, they demonstrate the urgent need to invest in structural support for the profession.

The main strength of our study is the richness of the data, which was collected by two professional Somali women with midwifery backgrounds, coupled with a collaborative approach throughout the research process between the research team in and outside of Somaliland. Research generated by midwives is needed to create evidence-informed policies to support the integration of the profession into health systems.[9 27 28] While there was much sharing during the FDGs, the setting may have limited how much individuals shared, as some may not have wanted to disclose personal details of their experiences in a group setting, especially given the sensitive topic areas.

## CONCLUSION

Maternal mortality and morbidity remain a significant challenge in Somaliland, especially in relation to services that are highly stigmatised, such as contraception, abortion care (including PAC) and SGBV. Midwives are trusted members of the community and are often the first point of care for such services. They truly are integral to providing women and girls with safe, equitable SRHR. Our research provides a rich portrayal of midwives who are essential to the provision of equitable SRHR, yet are not fully supported by policies, laws or institutions, often living in fear of the consequences of their behaviours. Our research highlights the importance of (1) the development and implementation of context-based written guidelines and policies that clearly define the role of midwives and provide them with clinical guidance, (2) investing in the professional development of midwives by providing training and supporting their efforts to provide SRHR services through the provision of necessary resources and equipment in Somaliland and similar settings and (3) the critical need for mental health support for midwives, especially those who have had traumatic experiences related to FGM/C. By addressing the challenges and controversies facing midwives, we can help to improve the quality of care provided to women and girls and ultimately enhance their reproductive health outcomes.

**Acknowledgements** The authors would like to thank the midwives who shared their personal lived experiences. Also, the authors would like to thank Dalarna University for providing the resources that were necessary for this research.

**Contributors** Conceptualisation of the study design and method, data interpretation and editing and proofreading: all authors. Planning and implementation of the study, including data collection: JAE, AE and RTM (online) supported by MK-A. Data analysis: RTM, KB, JAE and CM supported by MK-A. Writing: RTM, KB, CM and MK-A. RTM is responsible for the overall content as guarantor.

**Funding** This study was supported by Dalarna University with an award, Karolinska Institute with an award and Swedish Research Council with an award Dnr 2020-04372. The funders played no part in the conduct of this research, and the views expressed in this publication are those of the authors.

**Competing interests** None declared.

**Patient and public involvement** Patients and/or the public were not involved in the design, or conduct, or reporting, or dissemination plans of this research.

**Patient consent for publication** Not applicable.

**Ethics approval** This study involves human participants, and approval to conduct the study at the health facilities was obtained from the Somaliland Ministry of Health and Development, MOHD/DG: 2/J-cG/2020, and ethical clearance was approved by the ethical board at the University of Hargeisa, Somaliland, DRCS/28/5/2020. Participants gave informed consent to participate in the study before taking part.

**Provenance and peer review** Not commissioned; externally peer reviewed.

**Data availability statement** Data are available upon reasonable request.

and indication of whether changes were made. See: https://creativecommons.org/licenses/by/4.0/.

**ORCID iD**
Rahel Tesfa Maregn http://orcid.org/0000-0002-4449-7026

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
