## [Reviewer comments · BMJ Open]

ARTICLE DETAILS

TITLE (PROVISIONAL)	A qualitative study of the roles of midwives in the provision of sexual and reproductive health care services in the Somaliland health system.
AUTHORS	Maregn, Rahel Tesfa; Bourret, Kirsty; Egal, Jama Ali; Esse, Amina; Mattison, Cristina; Klingberg-allvin, marie

VERSION 1 – REVIEW

REVIEWER	Raven, Joanna Liverpool School of Tropical Medicine, Department of International Public Health
REVIEW RETURNED	04-Sep-2022

GENERAL COMMENTS	This is an interesting study on role of midwives in SRHR in Somaliland. It uses an exploratory qualitative design with FGDs with midwives to understand their perspectives of service provision. My main points to be addressed are: 1. Introduction: a stronger rationale for the study should be provided. What is the gap in evidence that this paper is contributing to? Why is this paper useful - how can these findings be used and by whom?2. Study sites selection: rationale why Hargeisa was selected as the study site – would there be differences in other areas of Somaliland – more rural areas may be more resource constrained and more traditional? Also are there any differences in the MCHs - size, workload, number of midwives, number of other staff. And any differences in findings across these facilities?3. The analysis section is also not clear - who did the analysis, how did you come up with the themes and sub themes, and the generic codes. One of the strengths highlighted was the collaborative approach of the research team and yet this does not strongly come through in the methods or analysis.4. This point is linked to point 3. The structure of the findings does not really work. There are some overlaps between the two different sections of results for example, FGM, the need for mental health support for midwives, challenging position midwives are in because of need to get male consent. Also the Somaliland context which includes socio cultural norms influences midwives roles and responsibilities -s so it difficult to separate the two themes as they are integrated. The structure of the results needs a rethink.5. The discussion should include some implications for policy and practice for Somaliland and similar settings.6. In the discussion you highlight UNFPA four key investment areas and that the findings provide important starting points to support midwives in Somaliland across all key areas. However, the findings do not really identify areas of workforce and midwifery governance
--

	and leadership – it would be good to make these points more explicit. 7. The abstract findings do not really reflect the findings of the paper. 8. There are some punctuation errors in the paper that should be addressed.
--	---

REVIEWER	Willott, Chris King's College London, King's Centre for Global Health and Health
REVIEW RETURNED	12-Dec-2022

GENERAL COMMENTS	This is a very good study, in which you reveal important information about the challenges faced by midwives in Somaliland as they navigate the relationships between cultural norms, the law and the safety and wellbeing of themselves and their patients. The main change I would recommend is to remove table 1 and replace with more narrative, supported by some of the quotes from the table. Specific comments are below.  • P4, line 27: Somaliland isn't sovereign; the terminology here needs to be correct. It would be better to comment that Somaliland is a de facto (but not de jure) independent state and functions as such. It is, however, not a sovereign state. • P4 line 27-28: remove words 'in a global comparison'. • Is there a potential issue with identification of the midwives, as the MCHs where they took place are named? Did all participants consent to this? • I'm not convinced that table 1 is the best way to present the evidence on the participants' views. It doesn't work structurally as the reader is given all the information at once, instead of the narrative explaining the situation and using quotes to support the narrative. The more standard approach of providing description interspersed with quotes (from p8 line 23 onwards) is much more effective • The term 'cultural law' is used repeatedly; as this is not a formal 'law', the term should be replaced with 'cultural norm' • P11, line 216, it is noted that 'pregnancy outside marriage is illegal'. Does this mean that sex outside marriage is illegal? Needs clarification • P13 line 39; you describe Somaliland as a post-conflict setting, but this has not been mentioned before, so it's unclear why it is included here. Conflict is not mentioned in any of the empirical data so I can't see the value of including this phrase. • P15, line 18; the last sentence needs to be reworded
--

VERSION 1 – AUTHOR RESPONSE

Reviewer 1

• This is an interesting study on role of midwives in SRHR in Somaliland. It uses an exploratory qualitative design with FGDs with midwives to understand their perspectives of service provision. Thank you very much for taking the time to review our manuscript.

1. Introduction: a stronger rationale for the study should be provided. What is the gap in evidence that this paper is contributing to? Why is this paper useful - how can these findings be used and by whom? Page 5, lines 5-18

We acknowledge the comment and thank the reviewer for that observation. In response, we have made adjustments to the revised manuscript to provide a strong rationale for the study and to clarify the gap that our paper aims to fill. We have also highlighted the potential uses and audiences for our

findings.

2. Study sites selection: rationale why Hargeisa was selected as the study site – would there be differences in other areas of Somaliland – more rural areas may be more resource constrained and more traditional? Also are there any differences in the MCHs - size, workload, number of midwives, number of other staff. And any differences in findings across these facilities? Page 5, lines 28-34
Page 12, lines 6-9

We acknowledge the comment and thank the reviewer for that observation. Because of word count only some of these points were included in the initial manuscript. In response, we have included in the manuscript description of the MCHs, including staff and population served.

Furthermore, in our study, we find differences in the availability and type of FP methods and PAC service across MCHs. These findings are briefly presented in the result section. These six MCHs were chosen for data collection because they were of similar size, number of midwives working as staff and also catered to a large IDP population. In the future, it would be good to also investigate this further outside of the capital city, but for this study, we wanted to focus on Hargeisa as there are more Midwives working in Hargeisa.

3. The analysis section is also not clear - who did the analysis, how did you come up with the themes and sub themes, and the generic codes. One of the strengths highlighted was the collaborative approach of the research team and yet this does not strongly come through in the methods or analysis.

Page 6, lines 9-17.

We acknowledge the comment and thank the reviewer for that observation. Though because of word count, this was briefly explained in the contributor's section of the initial manuscript, we acknowledge the need for a detailed explanation of the data analysis on the methodology. In response, we have provided a detailed explanation of the analysis process with more clarity in the revised version. In addition, we added a revised table showing the data extraction process.

4. This point is linked to point 3. The structure of the findings does not really work. There are some overlaps between the two different sections of results for example, FGM, the need for mental health support for midwives, challenging position midwives are in because of need to get male consent. Also, the Somaliland context which includes sociocultural norms influences midwives roles and responsibilities so it difficult to separate the two themes as they are integrated. The structure of the results needs a rethink.

Page 7-13

We acknowledge the comment and thank the reviewer for that observation. We have revised the structure of our findings and made adjustments to the result section.

5. The discussion should include some implications for policy and practice for Somaliland and similar settings.

Page 15, lines 23-31

We acknowledge the comment and thank the reviewer for that observation. The revised version now includes policy implications for Somaliland and similar settings.

6. In the discussion you highlight UNFPA four key investment areas and that the findings provide important starting points to support midwives in Somaliland across all key areas. However, the findings do not really identify areas of workforce and midwifery governance and leadership – it would be good to make these points more explicit.

Page 15, lines 23-31

We acknowledge the comment and thank the reviewer for that observation. We have included key investment areas identified in the policy implications.

7. The abstract findings do not really reflect the findings of the paper.

Page 2, lines 12-25

We acknowledge the comment and thank the reviewer for that observation. We have made adjustments to the abstract.

8. There are some punctuation errors in the paper that should be addressed

We acknowledge the comment and address the issue in the revised version.

Reviewer 2

1. This is a very good study, in which you reveal important information about the challenges faced by midwives in Somaliland as they navigate the relationships between cultural norms, the law and the safety and wellbeing of themselves and their patients. The main change I would recommend is to remove table 1 and replace with more narrative, supported by some of the quotes from the table.

Page 8, lines 32-35,

Page 9, lines 1-5, 25-27

Page 10, lines 1,2,11-18, 38,39

Page 12, lines9-13,22, 39-41

Page 13, lines 6-8,17,27,28

Thank you kindly for your time and interest in our work. We appreciate your insightful and very pertinent comments. We used a table with quotes to save words in the main text and to show the data extraction process. We acknowledge the comment, and in the revised document, we add quotes from tables to the narratives and a revised table to the analysis section.

2. P4, line 27: Somaliland isn't sovereign; the terminology here needs to be correct. It would be better to comment that Somaliland is a de facto (but not de jure) independent state and functions as such. It is, however, not a sovereign state.

Page 4, lines 21,22

We acknowledge the comment and thank the reviewer for that observation. In response, we have removed controversial political statements in order to increase the impact of our research.

3. P4 line 27-28: remove words 'in a global comparison'.

We acknowledge the comment and thank the reviewer for that observation. We have removed the word in the revised document.

4. Is there a potential issue with identification of the midwives, as the MCHs where they took place are named? Did all participants consent to this?

We thank the reviewer for that observation. We removed this information from the manuscript as it has ethical implications.

5. I'm not convinced that table 1 is the best way to present the evidence on the participants' views. It doesn't work structurally as the reader is given all the information at once, instead of the narrative explaining the situation and using quotes to support the narrative. The more standard approach of providing description interspersed with quotes (from p8 line 23 onwards) is much more effective.

Page 6-12

We acknowledge the comment and thank the reviewer for that observation. We used a table with quotes to save words in the main text and to show the data extraction process. In response, we have adjusted the result presentation accordingly.

6. The term 'cultural law' is used repeatedly; as this is not a formal 'law', the term should be replaced with 'cultural norm'.

Page 9, line 17, page 11, line 17, page 11, line 17

We acknowledge the comment and thank the reviewer for that observation. The term is changed to 'cultural norm' in the revised document.

7. P11, line 216, it is noted that 'pregnancy outside marriage is illegal'. Does this mean that sex outside marriage is illegal? Needs clarification

Page 10, lines 7-18

We acknowledge the comment and thank the reviewer for the observation. Yes, sex outside of marriage is illegal in the context, and that is what the participants were also referring to. We added more quotes from participants to support the narrative.

8 P13 line 39; you describe Somaliland as a post-conflict setting, but this has not been mentioned before, so it's unclear why it is included here. Conflict is not mentioned in any of the empirical data so I can't see the value of including this phrase.

We acknowledge the comment and thank the reviewer for that observation. In response, we have removed the phrase in the revised manuscript.

9. P15, line 18; the last sentence needs to be reworded.

Page 14, lines 29-33

We acknowledge the comment and thank the reviewer for that observation. We have made adjustments to the sentence.

Editor(s)' Comments

1. Please complete a thorough proofread of the text and correct any spelling and grammar errors that you identify.

Done

2. Please include a copy of the interview/discussion guide as a supplementary file or a link to where readers can access it.

Included

VERSION 2 – REVIEW

REVIEWER	Willott, Chris King's College London, King's Centre for Global Health and Health
REVIEW RETURNED	24-Feb-2023
GENERAL COMMENTS	All of the comments I made on the first draft of this paper have been addressed. I am happy that the manuscript should be accepted in its current form.